# Mental Health among Higher Education Students during the COVID-19 Pandemic: A Cross-Sectional Survey from Lithuania

**DOI:** 10.3390/ijerph182312737

**Published:** 2021-12-02

**Authors:** Emilijus Žilinskas, Giedrė Žulpaitė, Kristijonas Puteikis, Rima Viliūnienė

**Affiliations:** 1Faculty of Medicine, Vilnius University, 03101 Vilnius, Lithuania; emilijus.zilinskas@mf.stud.vu.lt (E.Ž.); giedre.zulpaite@mf.stud.vu.lt (G.Ž.); kristijonas.puteikis@mf.stud.vu.lt (K.P.); 2Clinic of Psychiatry, Vilnius University, 03101 Vilnius, Lithuania

**Keywords:** suicide, COVID-19, self-reported health, sense of coherence

## Abstract

Mental health issues—anxiety, depression, suicidal ideation and behavior—are prevalent among students of higher education. The COVID-19 pandemic further affected students’ daily life through academic and socioeconomic disturbances. We set out to investigate students’ mental health amidst the COVID-19 pandemic and conducted a cross-sectional online survey at higher education institutions in Lithuania in 2021. The questionnaire consisted of the Hospital Anxiety and Depression scale (HADS) and the Sense of Coherence scale (SOC-3) questionnaires, evaluation of suicidal risk, experiences during the COVID-19 pandemic and self-rated health status (SRHS). Among 1001 students who completed the survey, the prevalence of clinically relevant anxiety was high (46.6%) and contrasted with the lower prevalence of depression (11.1%). 37.5% of all students admitted that they had thought about suicide at least once during their lifetime and a similar number of students thought about suicide during the previous year. High levels of anxiety and depression were statistically significant predictors of suicidal ideation and planning during the past year in binary regression models. High SRHS (higher score refers to more positive health status) was the only significant independent variable associated with less frequent suicidal attempts in the past year (*p* < 0.01, OR = 0.29, 95% CI = 0.12 to 0.66). Our study highlights anxiety and suicidality to be burdensome mental health issues among higher education students in Lithuania during the COVID-19 pandemic.

## 1. Introduction

University and college students belong to a group of young adults who express high levels of mental health issues, such as depression and anxiety [1,2]. For example, a report by the WHO World Mental Health Surveys International College Student initiative estimated that the prevalence of mental disorders in a sample of nearly 14,000 students was around 30% [3]. However, students’ mental distress may be even higher: as depressive and anxious symptoms form a continuum and range from none to severe, students may express such levels of anxiety and depression that may not be considered to be clinically significant [4,5,6]. Mental distress (both clinically diagnosed and sub-syndromic factors) is considered to negatively influence academic performance and social well-being of youths (however, due to bidirectional relationships an opposite pathway is also possible) [7]. Furthermore, students’ mental distress is connected to suicidality [8]. Studies highlight that students of higher education are prone to suicidal ideation [9,10]. According to the WHO Global Health Estimates for the year 2019, suicide is the fourth leading cause of death in the population of people aged 15 to 29 years [11], a considerable part of whom study at universities and colleges.

High prevalence of mental health disturbances as well as suicidal ideation and behavior among studying young adults is mainly explained by difficulties related to transitioning into adulthood [12,13]. In addition to stress posed by academic pressure, students experience future-projected uncertainties related to later marriage, childbirth and a later start of their careers [14]. Finally, many students who express high levels of mental distress do not reach for mental health counseling services due to fear of discrimination, concerns about the cost of such services or even not knowing that such services exist [15,16,17].

The COVID-19 pandemic has also significantly affected the lives of higher education students. Alongside struggles in maintaining personal relationships due to general social distancing measures and restrictions, students faced difficulties in maintaining their attention during the learning process [18]. Moreover, students experienced financial instability due to lost on-campus jobs [19]. Disrupted research projects and internships impaired students’ competitiveness on the future job market [19]. Thus, young adults are considered to be one of the most vulnerable groups in terms of mental health issues during the COVID-19 crisis [20,21,22].

Various studies investigated factors (both modifiable and not) that may affect the impact that the pandemic has on students’ mental health—the most frequent variables assessed are female sex, worse living conditions, family income instability, lower level of social support, having relatives infected with COVID-19, a history of mental disorder etc. [23,24,25,26,27]. However, only a few studies focused on a person’s inner resources to cope with new problems related to the COVID-19 crisis. One of such resources is the sense of coherence. For example, it was suggested that high sense of coherence, i.e., the ability to perceive stressful situations as understandable, manageable and meaningful, may be beneficial during the COVID-19 pandemic [28]. Various studies supported such assumptions as higher sense of coherence may serve as a protective factor for mental health and has been associated with higher life satisfaction during the health crisis [28,29,30]. Similarly, a higher sense of coherence in university students was associated with better mental health and a health-promoting lifestyle, yet these studies were done before the pandemic [31,32]. However, studies that assess students’ sense of coherence and its impact on their mental health are lacking: to our knowledge, and only one study investigated students’ sense of coherence and its mediating effect on mental health during the pandemic [33].

The aims of our study were to (1) evaluate students’ levels of depression, anxiety and suicidal ideation and behavior during the COVID-19 pandemic, and (2) investigate how depressiveness, anxiety, pandemic-associated experiences and sense of coherence are related with students’ suicidality.

## 2. Materials and Methods

### 2.1. Participants

A total of 1001 students participated in this study. Participants were students from higher education institutions in Lithuania, i.e., universities and colleges. According to the data provided by the Official Statistics Portal of Lithuania, there were approximately 104,000 individuals studying at the universities and colleges in Lithuania in the academic year of 2020–2021 [34]. This indicates that around 1% of students from higher education institutions in Lithuania participated in our study. Characteristics of the respondents in our study are presented in Table 1.

### 2.2. Design and Procedure

We used a non-probability (convenience and snowball) sampling technique by distributing an online questionnaire from 31 January to 7 February 2021 during the second local wave of COVID-19 and a national lockdown in Lithuania. The online survey was distributed via social media groups, such as: “Medical students from X university/college”, “All students from X university/college”, “Students of Economics 2019”, “All students from X city” and similar. We searched for existing groups based on a list of institutions of higher education in Lithuania; thus, students from different educational centers could have participated in the study [35]. The number of members in these groups ranged from a dozen to more than 1000. We reached out to all social groups of students found in social media platforms. We did not use any other way to reach students. Students were asked to complete the online anonymous survey and to share the questionnaire with their acquaintances through various social media channels. The latter technique was used to increase the reach and sample size of our survey. The survey took approximately 10 minutes to complete. We did not use any inclusion or exclusion criteria for our respondents, thus, every student who saw the invitation to complete the survey could have participated. An exception was exchange and foreign students who were unable to complete the questionnaire in Lithuanian. Because of the snowball sampling technique used to collect information, we could not state the response rate of the survey. 

### 2.3. Instruments

The assessment protocol consisted of questions about demographic and study-related characteristics (sex, age, way of living during studies, study field and year of studies). Furthermore, students completed the Lithuanian version of the Hospital Anxiety and Depression Scale (HADS) [36] and the Sense of Coherence (SOC-3) scale [37]. Respondents were also asked to evaluate their self-perceived health status and provide information about suicidal ideation and behavior. Lastly, respondents evaluated their experiences related to the COVID-19 crisis. The questionnaire form used in the study is provided as Appendix A.

#### 2.3.1. The Hospital Anxiety and Depression Scale (HADS)

The HADS is used to screen for the presence of symptoms of depression and anxiety [36]. It has been developed with the intention for a simple and rapid administration in everyday settings and confirmed to be valid and reliable in various populations [38,39]. This instrument contains 14 questions: seven items assess depressiveness, and another seven items express the level of anxiety. Scores for items in each subscale of the HADS were summed to produce an Anxiety score (HADS-A) or a Depression score (HADS-D). Each item is rated on a 4-point scale (higher scores indicate more expressed symptoms) with a total score ranging from 0–21 for each subscale. We used the original cut-off scores (≥11) proposed by the authors of the HADS for detecting cases of anxiety and depression that seem to be adequate [38,40]. The subscale scores can be added to produce a total score of the HADS. In the current study, the reliability of the scale was acceptable (HADS-A: Cronbach’s α = 0.84, McDonald’s ω = 0.85, HADS-D: Cronbach’s α = 0.74, McDonald’s ω = 0.74, HADS (total): Cronbach’s α = 0.86, McDonald’s ω = 0.86).

#### 2.3.2. The Sense of Coherence (SOC-3) Scale

The SOC-3 scale is a simplified version of the SOC-13 and SOC-29 scales and was selected instead of the latter two versions because of its brevity. The SOC-3 questionnaire consists of three questions concerning manageability, meaningfulness and comprehensibility—factors representing one’s internal resources when coping with difficulties in life [41,42]. The questions were: “Do you usually see a solution to problems and difficulties that other people find hopeless?” (manageability), “Do you usually feel that your daily life is a source of personal satisfaction?” (meaningfulness), “Do you usually feel that the things that happen to you in your daily life are hard to understand?” (comprehensibility). Each item is rated on a scale from 0 to 2 and the possible answers are: “Yes, usually” (0 points), “Yes, sometimes” (1 point) or “No” (2 points) (points scored in a question regarding comprehensibility were reversed). A greater SOC-3 score indicates worse sense of coherence. This instrument was included to investigate whether an intrinsic mechanism of resilience may be related to mental health and self-reported issues during the COVID-19 pandemic. In our study, the scale had optimal reliability (Cronbach’s α = 0.84, McDonald’s ω = 0.85).

#### 2.3.3. Self-Reported Health Status (SRHS)

Respondents were asked to evaluate their own health status. The answer is rated on the Likert scale, where 1 indicates worst imaginable health and 5 reflects best imaginable health. As SRHS has predictive value on future mortality [43,44], we hypothesized that this item may have a significant relationship with students’ suicide attempts.

#### 2.3.4. Suicidal Ideation and Behavior

We did not utilize any suicide risk assessment scales due to their questionable usefulness in terms of sensitivity and specificity and lack of options standardized for the Lithuanian population [45,46]. Therefore, questions relating to suicidality were created ad hoc to include a primary question for initial screening and then provide information on suicidal ideation or behavior in the past 12 months.

Firstly, students reported whether they thought about committing suicide, planned or tried to commit suicide at any time during their lifetime. If affirmative, they were considered to have risk for suicide and had to answer four other questions concerning symptoms of sadness and apathy as well as suicidal ideation and behavior during the past year.

The latter questions were:During the last 12 months, have you experienced sadness from day to day for at least 2 weeks and did not want to do anything?During the last 12 months, have you thought of committing suicide?During the last 12 months, have you engaged in creating suicidal plans?During the last 12 months, have you tried to commit suicide?

#### 2.3.5. Experiences Related to the COVID-19

Finally, students had to evaluate their experiences during the COVID-19 pandemic. Nine statements were proposed by the authors: (1) “My physical health got worse”, (2) “I felt more anxiety than usually”, (3) “I felt more sadness than usually”, (4) “My academic performance got worse”, (5) “I had no comfortable place to study”, (6) “I had difficulties focusing on my studies”, (7) “My career prospects got worse”, (8) “My personal relationships got worse”, (9) “My income became lower”. Each statement was rated on a 5-item Likert scale, where 1 point means total disagreement and 5 points means total agreement. We did not utilize any validated assessment tools to evaluate pandemic-associated changes in students’ life because of a lack of such instruments in the scientific community at that time. Therefore, statements were considered ad hoc to investigate three main domains of interest: (1) changes in mental health, (2) changes in professional (student) activities and (3) socioeconomic consequences of the pandemic. If combined, the reliability of these questions was adequate (Cronbach’s α = 0.80, McDonald’s ω = 0.80) and an exploratory factor analysis suggested that the three domains explained 63.4% of the variance (Kaiser–Meyer–Olkin measure = 0.827) (Table 2).

Students were also asked about their and their families’ COVID-19 infections and hospitalizations.

### 2.4. Ethics

According to local regulations, the survey did not require approval from a bioethics committee as it was completely anonymous (no personal data were collected) and completed voluntarily via internet.

### 2.5. Data Analysis

Data were analyzed using Microsoft Excel v16 and IBM SPSS v26. The minimum required sample size for our study was calculated in G*Power for a one-way ANOVA of six groups (based on the field of study, i.e., biomedical, physical sciences, social sciences, humanities, art, technologies) with α = 0.05, 1-β = 0.95 and f = 0.25 and was *n* = 324.

The reliability of the instruments used was investigated by calculating Cronbach’s α and McDonald’s ω. The dimensionality of COVID-19-related questions was assessed by performing an exploratory factor (principal component) analysis with an oblique factor rotation (factor loadings > 0.40 were included in each factor).

The normality of variable distribution was assessed by the Kolmogorov–Smirnov test. Mann–Whitney U, Kruskal–Wallis (H) tests were used for between-group (e.g., male vs. female, students from different fields of study) comparison. Spearman’s correlation was used for correlation analysis (between COVID-19-related items, HADS, SOC-3 and SRHS) and for data of non-normal distribution as well as single-item Likert scale ordinal data. The Student’s *t*-test or the one-way ANOVA were employed for normally-distributed data with equality of variances. Binary logistic regression modeling was used to estimate variables that are associated with suicidality. Results were treated as statistically significant if *p* < 0.05.

## 3. Results

There were 466 (46.6%) and 111 (11.1%) participants whose symptoms of anxiety and depression were evaluated to be above the cut-off score (HADS-A/D ≥ 11) for clinical relevance, respectively. Very weak to medium correlations were present between the HADS, SOC-3, self-reported health and COVID-19-related questions (Table 3). SOC-3 also correlated with HADS-A (r_s_ = 0.55, *p* < 0.001), HADS-D (r_s_ = 0.55, *p* < 0.001) and SRHS (r_s_ = −0.43, *p* < 0.001).

Female individuals had higher levels of anxiety (M_female_ = 10.6 ± 4.3 vs. M_male_ = 8.7 ± 4.3, Z = −5.72, *p* < 0.001) and worse sense of coherence (M_female_ = 2.7 ± 1.3 vs. M_male_ = 2.4 ± 1.3, Z = −3.10, *p* = 0.002). They more frequently reported worse physical health (M_female_ = 3.5 ± 1.2 vs. M_male_ = 3.2 ± 1.3, Z = −2.01, *p* = 0.045), increased anxiety (M_female_ = 3.7 ± 1.6 vs. M_male_ = 3.3 ± 1.2, Z = −4.09, *p* < 0.001) and depressiveness (M_female_ = 3.7 ± 1.2 vs. M_male_ = 3.4 ± 1.2, Z = −3.92, *p* < 0.001) as consequences of the COVID-19 pandemic. Individuals whose family members were hospitalized because of COVID-19 (*n* = 39) more often indicated a worsening in academic performance, compared to those who did not have hospitalized relatives (M_hospitalized_ = 3.2 ± 1.4 vs. M_not hospitalized_ = 2.8 ± 1.3, Z = −2.06, *p* = 0.039).

Symptoms of anxiety were more expressed among students of humanities as opposed to those of physical or biomedical sciences (H(5) = 24.89, *p* < 0.001, post hoc *p* < 0.001 and *p* = 0.001, respectively). Students of the social sciences less frequently agreed that their study results became worse in comparison to students of biomedical fields (H(5) = 15.31, *p* = 0.009). The latter more often reported that they expect worse career prospects because of the pandemic (H(5) = 15.51, *p* = 0.008). Art students more frequently had no comfortable place to study (H(5) = 14.34, *p* = 0.014).

Neither SOC-3 nor HADS-A/D scores depended on the students’ way of living. However, participants who lived with their partners were less likely to agree that the pandemic negatively influenced their interpersonal relationships (H(3) = 32.44, *p* < 0.001).

There were no statistically significant findings relating suicidality-associated questions with the type of living or study fields, except for a higher frequency of general suicidality among art and humanities students (60.5% and 69.7% vs. 49.1–57.6% in other fields, χ^2^ = 14.46, *p* = 0.013).

Binary regression modeling revealed that results of the HADS-D and HADS-A scales are significantly associated with general suicidality, suicidal ideation and suicidal planning, but only self-reported health was related to suicidal attempts (Table 4).

## 4. Discussion

We observed statistically significant associations between the HADS-A/D scores and the students’ suicidal ideation and planning in the past year. The latter finding was expected, as studies indicate a close relationship between mental health and suicidal ideation and behavior [47,48]. Self-rated health status (SRHS) was another variable related to students’ general and past-year suicidality—SRHS was also the only significant predictive factor for students’ past-year suicide attempt. To the best of our knowledge, only one study revealed similar connections between self-rated health status and suicidality in young individuals [49]. Thus, our study provides additional information that self-rated health status may be a predictive factor for suicidal behavior among youth. However, the measurement of SRHS is determined by various physical, psychological as well as socio-demographic aspects [50,51]. Therefore, it is difficult to clarify specific variables that may be the most relevant determinants of worse SRHS. We could only speculate that one of such factors may be a reduction of students’ physical activity due to the pandemic, as higher physical activity is related both to better self-rated health and lower suicidal ideation [52,53,54].

As expected, students with higher inner resources of resilience (defined as sense of coherence in our study) expressed lower anxiety and depression rates. Furthermore, they were less likely to have ever thought about suicide during their lifetime. The latter finding is consistent with data from other studies that highlight a close connection between worse SOC and higher rates of suicidal ideation and behavior [55,56,57,58,59]. However, SOC was not significantly associated with students’ suicidality in the past year (i.e., since the start of the COVID-19 pandemic). The reason for this remains unclear—other factors may be more important than SOC in the context of the COVID-19 health crisis. SOC was lower among females and this finding is consistent with data from another study [60], but findings remain contrasting [61,62]. Females are more vulnerable to mental distress during the pandemic [63,64,65]. Therefore, it would be reasonable to focus more on improving female students’ inner resilient resources to efficiently cope with pandemic-associated mental disturbances.

During the COVID-19 pandemic, many studies investigated mental health of medical students as they typically show elevated levels of anxiety [66]. However, in contrast to the general view, we did not observe any significant differences in terms of depression, anxiety as well as suicidal ideation and behavior between medical students and students from other specialties. The Student Experience in the Research University (SERU) Consortium survey revealed similar results, as students of health sciences were not among those with highest prevalence of anxiety and depression [67]. The only difference observed in our study was the experience of worse career prospects due to the pandemic—it may be reasonable as medical students faced additional challenges related to remote studying and lack of ability to improve their clinical skills. However, our data highlight the importance to assess the mental health of all students of higher education equally rather than focus on one specialty.

We have also analyzed students’ experiences related to the pandemic through changes in their mental health and activities as well as socioeconomic consequences and found weak to moderate correlations between the latter experiences and students’ levels of sense of coherence, anxiety, depression and perceived health status. However, our list of possible pandemic-associated changes that may affect students’ mental health was not exhaustive. For example, studies highlighted negative changes in dietary profile among adolescents and students of higher education during the current pandemic [68,69]. Knowing the possible bidirectional relationship between mental health and diet, it would be useful to include questions related to students’ diet in future research [68,70]. Further, evidence suggests that students’ lifestyle during the pandemic became more sedentary [71,72]. As with diet, physical activity changes should also be monitored due to its close relationship with mental health [73,74]. Moreover, students of higher education consumed alcohol in greater extent during the pandemic [75]; it may serve as another risk factor for students’ mental health outcomes, as alcohol consumption is related both to anxiety and depression [76]. Lastly, one cross-sectional study from China explored college students’ mental health after re-opening of schools and highlighted personal as well as school regulations-related factors significant for mental health outcomes, such as fear of being infected, attitude toward COVID-19, self-quarantine, quarantine of classmates, routine assessment of temperature, wearing masks [77]. Thus, it is obvious that mental health disturbances of students from higher education will not disappear shortly after re-opening of colleges and universities and that the latter institutions should implement rational measures to minimize the negative impact of the current pandemic to students’ mental health (for example, by delivering clear and informative messaging to students, prioritizing and expanding student support services etc.) [78].

Our study indicates that students of higher education in Lithuania suffer from anxiety—almost one in two students (46.6%) expressed anxiousness that may be treated as clinically relevant. High prevalence of students’ anxiety has also been measured in other studies from Lithuania [79,80]. However, international data provide a significantly lower prevalence rates of anxiety among students. For example, a meta-analysis indicates the rate of anxiety among college students is 29% during the pandemic [81]; the rates of anxiety are also lower in individual studies [82,83,84,85]. We believe that such discrepancies between results from our study and current data may be explained by different instruments used to screen for students’ symptoms of anxiety. For instance, we employed the HADS, which is a short, easy-to-use instrument and has been recognized as a reliable tool to detect depressiveness and anxiety in different settings (e.g., adolescent studies) [86]. However, its usefulness for screening university and college students has not been widely explored. One study suggested that the HADS questionnaire may be prone to overestimate anxiety [87]. This may explain the high rate of anxiety as many before cited studies used the Generalized Anxiety Disorder scale-7 (GAD-7) questionnaire to detect the disorder. In contrast to anxiety, we observed a relatively low prevalence of depressive symptoms reaching the cut-off value for clinical relevance among students of higher education in Lithuania (11.1%, in contrast to the general estimate of 37% among college students during the pandemic) [81]. 

We found that more than a half of all students thought about suicide at least once during their lifetime. Furthermore, the prevalence of attempted suicide during students’ lifetime (4.4%) was higher compared to data from meta-analysis in the students’ population (2.7%) and from a large study that assessed adults (also 2.7%) [88,89]. We consider such measurements as a reflection of the poor general situation regarding suicide in Lithuania: according to the WHO, Lithuania ranks seventh among countries with the highest suicide rates in the world and first in Europe in 2019 [90]. This may be associated with significant socioeconomic shifts in the country during the last decades: Lithuanian society was emancipated from the Soviet regime and witnessed an economic boom until the global financial crisis in 2008 while having no effective national suicide prevention strategy [91]. Despite recent improvements in mental health services, it may be reasonable to consider that students in Lithuania are highly exposed to suicidal behavior among peers, relatives as well as the general society, and the latter aspect seems to be predictive for increased suicidality [92,93]. 

In terms of suicidal ideation in the past year (i.e., since national lockdown measures were first applied in Lithuania), 35.7% of all respondents admitted that they have thought about suicide. Such estimate was two times higher compared to the data from a systematic review on the prevalence of suicidal ideation and thoughts among university students during the current pandemic (17.8%) [94]. Such high estimates may be linked to general suicidality; it remains unclear, however, to what extent did the pandemic specifically affect students’ suicidal ideation and behavior. Previous research indicate pandemic-related stressors such as impaired family functioning, loneliness, and burden due to staying at home to be strongly related to suicidal ideation [95,96]. Our study, however, did not highlight any of the pandemic-associated variables to be significantly predictive for suicidal ideation, planning or attempt during the last year.

### Limitations

The cross-sectional design of the study did not allow us to measure causal relationships between the pandemic-associated changes and students’ mental health. Furthermore, all data were self-reported rather than collected by specialists in mental health through interviews or other means of objective evaluation. The intensity or risk factors of suicidal ideation and behavior were not assessed as well, thus, the interpretation of suicidal risk in our study should be cautious. Non-respondent bias may have also favored the inclusion of students who already suffer from mental health issues or are especially introspective towards their mental health and exclusion of healthy participants. This fact as well as a non-probability sampling technique decreased the external validity of our study. 

## 5. Conclusions

Our study provides further evidence that mental health issues were prevalent among students of higher education during the COVID-19 pandemic. Symptoms of anxiety were more prevalent than depressive symptomatology in our study. However, we report high rates of suicidal ideation. While the HADS was associated with suicidal ideation and planning, only the self-reported health status was related to previous suicidal attempts. Measuring sense of coherence may be relevant when considering suicidal ideation and behavior during one’s lifetime, but it was not found to be associated with these issues in the context of the pandemic.

## Figures and Tables

**Table 1 ijerph-18-12737-t001:** General characteristics of the participants in our study.

Characteristic	Categories	
Number of respondents		1001
Age	Median, range	20 (18–69)
	Mean, SD	20.8 (2.8)
Sex (*n*,%)	Male	225 (22.5)
	Female	776 (77.5)
Study field (*n*,%)	Biomedical sciences	330 (32.9)
	Physical sciences	131 (13.1)
	Humanities	89 (8.9)
	Art	114 (11.4)
	Social sciences	231 (23.1)
	Technologies	106 (10.6)
Way of living during studies (*n*,%)	Alone	142 (14.2)
	With family members	463 (46.2)
	With other students or peers	249 (24.9)
	With a partner	147 (14.7)
SOC-3	Median, range	3 (0–6)
	Mean, SD	2.6 (1.3)
HADS	Total score—median, range	16 (0–42)
	Total score—mean, SD	16.2 (7.0)
	HADS-D—median, range	6 (0–21)
	HADS-D—mean, SD	6.1 (3.6)
	HADS-A—median, range	10 (0–21)
	HADS-A—mean, SD	10.2 (4.4)
SRHS	Median, range	3 (1–5)
General suicidal risk (*n*,%)	Had thoughts of committing suicide Was engaged in creating suicidal plans Tried to commit suicide	375 (37.5) 140 (14.0) 44 (4.4)
Suicidal risk for the last 12 months (*n*, % of those who ever thought about committing suicide)	Experienced sadness from day to day for at least 2 weeks and did not want to do anything	387 (69.2)
	Had thoughts of committing suicide	358 (64.0)
	Was engaged in creating suicidal plans	98 (17.5)
	Tried to commit suicide	17 (3.0)
Has been infected with SARS-CoV-2 (*n*,%)		117 (11.7)
Was hospitalized because of COVID-19 (*n*,%)		4 (0.4)
A family member has been infected with SARS-CoV-2 (*n*,%)		295 (29.5)
A family member was hospitalized because of COVID-19 (*n*,%)		39 (3.9)

SOC-3—Sense of Coherence scale-3, HADS-A—the Hospital Anxiety and Depression Scale (anxiety subscale), HADS-D—the Hospital Anxiety and Depression Scale (depression subscale), SRHS—self-reported health status.

**Table 2 ijerph-18-12737-t002:** Results of an exploratory factor analysis (principal component analysis) after an oblique rotation (factor loadings > 0.40 are in bold) when questions related to the consequences of the COVID-19 pandemic were considered.

	Factor		
	1 (“mental health”)	2 (“studies”)	3 (“socioeconomic”)
Variance explained	39.4%	12.7%	11.3%
**COVID-19-related variable**			
More sadness	**0.844**	0.079	−0.003
More anxiety	**0.843**	0.034	−0.017
Worse physical health	**0.694**	0.062	−0.034
Worse study results	0.035	**0.833**	−0.079
Difficulties to concentrate	0.198	**0.741**	0.002
No comfortable place to study	−0.068	**0.677**	0.211
Lower income	−0.078	0.038	**0.837**
Worse career prospects	0.042	0.149	**0.702**
Worse interpersonal relationships	**0.468**	−0.160	**0.473**

**Table 3 ijerph-18-12737-t003:** Spearman’s correlation coefficients between questions related to the COVID-19 pandemic, SOC-3, HADS scales and self-reported health.

COVID-19-Related Statement (a)	SOC-3	HADS-D	HADS-A	SRHS
Worse physical health	0.21 **	0.35 **	0.29 **	−0.45 **
More anxiety	0.27 **	0.33 **	0.47 **	−0.31 **
More sadness	0.31 **	0.39 **	0.41 **	−0.34 **
Worse study results	0.17 **	0.22 **	0.13 **	−0.19 **
No comfortable place to study	0.19 **	0.24 **	0.20 **	−0.18 **
Difficulties to concentrate	0.15 **	0.25 **	0.17 **	−0.21 **
Worse career prospects	0.14 **	0.24 **	0.19 **	−0.20 **
Worse interpersonal relationships	0.26 **	0.32 **	0.33 **	−0.25 **
Lower income	0.10 *	0.20 **	0.17 **	−0.15 **

a—measured as agreement to the statement on a scale from 1 (total disagreement) to 5 (total agreement). SOC-3—Sense of Coherence scale-3, HADS-A—the Hospital Anxiety and Depression scale (anxiety subscale), HADS-D—the Hospital Anxiety and Depression Scale (depression subscale), SRHS—self-reported health status, * *p* < 0.01, ** *p* < 0.001.

**Table 4 ijerph-18-12737-t004:** Binary logistic regression models for different suicide risk-related outcomes. Only respondents who ever had suicidal thoughts, plans or attempts (positive for general suicidal risk) were included in models 2 to 4.

	**Model 1: Dependent Variable: General Risk of Suicidality (*n* = 1001) (a)**			**Model 2: Dependent Variable: Suicidal Ideation in the Last 12 Months (*n* = 559)**		
**Independent Variable**	**β**	**Wald χ^2^**	**OR (95% CI)**	**β**	**Wald χ^2^**	**OR (95% CI)**
Intercept	−0.218	0.110	0.80 (0.22 to 2.93)	−1.660	3.977 *	0.19 (0.04 to 0.97)
Sex (b)	0.130	0.528	1.14 (0.80 to 1.61)	−0.006	0.001	0.99 (0.62 to 1.59)
SOC-3	0.284	16.294 ***	1.33 (1.16 to 1.53)	0.124	1.776	1.13 (0.94 to 1.36)
SRHS	−0.274	5.953 *	0.76 (0.61 to 0.95)	0.112	0.652	1.12 (0.85 to 1.47)
HADS-A	0.145	35.095 ***	1.16 (1.10 to 1.21)	0.062	4.367 *	1.06 (1.00 to 1.13)
HADS-D	0.088	8.744 **	1.09 (1.03 to 1.16)	0.134	12.136 ***	1.14 (1.06 to 1.23)
Worse physical health (C-19)	−0.098	1.790	0.91 (0.79 to 1.05)	−0.183	3.996 *	0.83 (0.70 to 1.00)
More anxiety (C-19)	−0.055	0.383	0.95 (0.80 to 1.13)	0.032	0.073	1.03 (0.82 to 1.30)
More sadness (C-19)	−0.095	1.082	0.91 (0.76 to 1.09)	0.008	0.004	1.01 (0.79 to 1.28)
Worse study results (C-19)	−0.084	1.490	0.92 (0.80 to 1.05)	0.064	0.535	1.07 (0.9 to 1.26)
No comfortable place to study (C-19)	−0.084	1.582	0.92 (0.81 to 1.05)	0.015	0.031	1.02 (0.86 to 1.20)
Difficulties to concentrate (C-19)	0.072	0.912	1.08 (0.93 to 1.25)	0.107	1.222	1.11 (0.92 to 1.34)
Worse career prospects (C-19)	−0.145	4.388 *	0.87 (0.76 to 0.99)	−0.001	<0.001	1.00 (0.84 to 1.18)
Worse interpersonal relationships (C-19)	0.041	0.396	1.04 (0.92 to 1.18)	0.064	0.541	1.07 (0.90 to 1.26)
Lower income (C-19)	0.023	0.137	1.02 (0.91 to 1.16)	−0.151	3.439	0.86 (0.73 to 1.01)
	**Model 3: Dependent Variable: Creation of a Suicidal Plan in the Last 12 Months (*n* = 559)**			**Model 4: Dependent Variable: A Suicidal Attempt in the Last 12 Months (*n* = 559)**		
**Independent Variable**	**β**	**Wald χ^2^**	**OR (95% CI)**	**β**	**Wald χ^2^**	**OR (95% CI)**
Intercept	−1.757	3.009	0.17 (0.02 to 1.26)	−0.026	<0.001	0.98 (0.01 to 88.17)
Sex (b)	0.297	1.052	1.35 (0.76 to 2.38)	−0.316	0.220	0.73 (0.20 to 2.72)
SOC-3	−0.040	0.118	0.96 (0.76 to 1.21)	−0.361	1.871	0.70 (0.42 to 1.17)
SRHS	−0.230	1.926	0.79 (0.57 to 1.10)	−1.253	8.706 **	0.29 (0.12 to 0.66)
HADS-A	0.067	3.508	1.07 (1.00 to 1.15)	0.005	0.004	1.01 (0.85 to 1.19)
HADS-D	0.116	7.402 **	1.12 (1.03 to 1.22)	0.151	2.914	1.16 (0.98 to 1.38)
Worse physical health (C-19)	−0.204	3.326	0.82 (0.66 to 1.02)	−0.380	2.227	0.68 (0.42 to 1.13)
More anxiety (C-19)	0.116	0.570	1.12 (0.83 to 1.52)	0.573	2.057	1.77 (0.81 to 3.88)
More sadness (C-19)	−0.073	0.216	0.93 (0.68 to 1.26)	−0.463	1.418	0.63 (0.29 to 1.35)
Worse study results (C-19)	0.104	0.940	1.11 (0.90 to 1.37)	0.236	0.977	1.27 (0.79 to 2.02)
No comfortable place to study (C-19)	0.139	1.722	1.15 (0.93 to 1.41)	−0.001	<0.001	1.00 (0.64 to 1.57)
Difficulties to concentrate (C-19)	−0.086	0.497	0.92 (0.72 to 1.17)	0.080	0.071	1.08 (0.60 to 1.95)
Worse career prospects (C-19)	−0.084	0.592	0.92 (0.74 to 1.14)	−0.342	1.893	0.71 (0.44 to 1.16)
Worse interpersonal relationships (C-19)	−0.117	1.227	0.89 (0.72 to 1.09)	0.056	0.053	1.06 (0.65 to 1.71)
Lower income (C-19)	0.032	0.102	1.03 (0.85 to 1.26)	0.157	0.476	1.17 (0.75 to 1.82)

a—Ever thought about committing suicide, planned or tried to commit suicide, b—0 = Male, 1 = Female, C-19—statements regarding COVID-19-related changes measured as agreement to the statement on a scale from 1 (total disagreement) to 5 (total agreement), SOC-3—Sense of Coherence scale-3, HADS-A—the Hospital Anxiety and Depression Scale (anxiety subscale), HADS-D—the Hospital Anxiety and Depression scale (depression subscale), SRHS—self-reported health status, Model 1: χ(14) = 228.28, *p* < 0.001, Model 2:, χ(14) = 63.35, *p* < 0.001, Model 3: χ(14) = 44.72, *p* < 0.001, Model 4: χ(14) = 35.35, *p* = 0.001. * *p* < 0.05, ** *p* < 0.01, *** *p* < 0.001.

## Data Availability

Raw data is available from the authors upon reasonable request.

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
