# Peer review of "Mental Health among Higher Education Students during the COVID-19 Pandemic: A Cross-Sectional Survey from Lithuania"

_ijerph, 2021, doi:10.3390/ijerph182312737_

Round 1

Reviewer 1 Report

Major comments

It is unclear which scale was used to evaluated suicidal risk, as HADS scale evaluates anxiety and depression, while SOC scale evaluates  sense of coherence. If authors deemed worse scores in both scales equal to suicidal risk, it is incorrect.

Minor comments

Sample size calculation is lacking. The quality of Figure 1 should be improved. Each table should comprise a footnote with abbreviations. Limitations should include the fact that all data are self-reported and that the sample size was not calculated prior the study. The study explores many factors, yet I feel that exploring diet or at least mentioning the dietary factors is needed (especially when considering that diet is bidirectionally linked to mental health). Notably, during the Covid outbreak, in adolescent population changed both lifestyle and dietary factors (PMID: 33975495). I believe this issue should be addressed in the discussion. ach table should comprise a footnote with abbreviations.

Author Response

Reviewer 1

We thank the reviewer for dedicating time to review our manuscript. We present our answers below.

It is unclear which scale was used to evaluated suicidal risk, as HADS scale evaluates anxiety and depression, while SOC scale evaluates sense of coherence. If authors deemed worse scores in both scales equal to suicidal risk, it is incorrect.

We agree that this point was unclear, and that suicidal risk cannot be evaluated by the HADS and SOC-3 questionnaires. We now state that suicidal risk was assessed by questions relating to suicidal ideation, planning and attempts directly:

2.3.4. Suicidal ideation and behavior

We did not utilize any suicide risk assessment scale due to questionable usefulness in terms of sensitivity and specificity and lack of options standardized for the Lithuanian population [24,25]. Therefore, questions relating to suicidality were created ad hoc to include a primary question for initial screening and then provide information on suicidal ideation or behavior in the past 12 months.

Firstly, students reported whether they thought about committing suicide, planned or tried to commit suicide any time during their lifetime. If affirmative, they were considered to have some risk for suicide and had to answer four other questions concerning suicidal ideations and behavior during the past year.

The latter questions were:

  • During the last 12 months, have you experienced sadness from day to day for at least 2 weeks and did not want to do anything?
  • During the last 12 months, have you thought of committing suicide?
  • During the last 12 months, have you engaged in creating suicidal plans?
  • During the last 12 months, have you tried to commit suicide?

Please note that the questionnaire is now available as a supplement to the manuscript.

Minor comments

Sample size calculation is lacking.

Sample size calculation is now provided in the Methods section:

The minimum required sample size for our study was calculated in G*Power for a one-way ANOVA of six groups (based on field of study) with α=0.05, 1-β=0.95 and f=0.25 and was n=324

The quality of Figure 1 should be improved.

Figure 1 has been removed from the manuscript, as suggested by Reviewer 4.

Each table should comprise a footnote with abbreviations.

We agree. Each table now has a footnote with abbreviations.

Limitations should include the fact that all data are self-reported and that the sample size was not calculated prior the study.

We agree that limitations should acknowledge that all data was self-reported. A respective sentence has been included in Limitations:

Further, all data was self-reported rather than collected by specialists in mental health through interviews or other means of objective evaluation.

A minimum sample size of 300-350 individuals was calculated and is now mentioned in the Methods section (it has been largely exceeded during the study):

The minimum required sample size for our study was calculated in G*Power for a one-way ANOVA of six groups (based on field of study) with α=0.05, 1-β=0.95 and f=0.25 and was n=324

The study explores many factors, yet I feel that exploring diet or at least mentioning the dietary factors is needed (especially when considering that diet is bidirectionally linked to mental health). Notably, during the Covid outbreak, in adolescent population changed both lifestyle and dietary factors (PMID: 33975495). I believe this issue should be addressed in the discussion. Each table should comprise a footnote with abbreviations.

We agree. While we did not include respective questions in the survey, but dietary factors should be mentioned as one of the factors associated with changes in mental health during the pandemic. A paragraph is now added in the discussion. Each table now has a footnote with abbreviations

However, our list of possible pandemic-associated changes that may affect students‘ mental health was not exhaustive. For example, studies highlighted negative changes in dietary profile among adolescents and students of higher education during the current pandemic [73,74]. Knowing the possible bidirectional relationship between mental health and diet, it would be useful to include questions related to students‘ diet in future research [73,75]. Further, evidence suggests that students‘ lifestyle during the pandemic became more sedentary [76,77]. As with diet, physical activity changes should also be monitored due to its close relationship with mental health [78,79]. Lastly, students of higher education consumed alcohol in greater extent during the pandemic [80]; it may serve as another risk factor for students‘ mental health outcomes, as alcohol consumption is related both to anxiety and depression [81].

Reviewer 2 Report

Thank you for the opportunity to review the manuscript entitled, "The comparison of students‘ anxiety, depression, suicidal risk and self-reported health before and during the COVID-19 pandemic”.

I believe this study investigated a topic relevant to the readers of “IJERPH”. The analysis of mental health among students of higher education who may be at risk of emotional or behavioral disorders has become a major focal point for researchers on a global scale, and it is now a priority for public health policies… The comparison of students‘ anxiety, depression, suicidal risk and self-reported health before and during the COVID-19 pandemic would significantly aid the proposals of improvement related to psychological orientation and counselling in this context, and it would also allow the definition of proceedings aimed to manage stress and anxiety in universities, especially in situations of crisis.

My opinion is that in the current form the manuscript does not have scientific quality to be published in IJERPH.

Structure of manuscript:

The authors should consider in your work the sections “Participants”, “Instruments”, “Procedure” and “Design”.

Introduction:

The introduction is very simply, does not analyze in detail the variables under study and the relationship between variables. Neither is the variable “Sense of Coherence” analyzed. The introduction lacks clear objectives. The discussion seems to contain information (studies) that would have been more appropriate in the Introduction.

The introduction must be rewritten.

Participants:

The participants were not randomly assigned. The number of participants from various higher education institutions in Lithuania is not representative and directly affects external validity. The higher education institutions in Lithuania was chosen at random? The authors should explain more clearly how the participants have been selected for the study.

The general characteristics of the sample need be addressed in the material and methods section (participants).

Instruments:

In first place, the instruments must always display two important qualities: reliability and validity. Is need the Cronbach's alpha (α) and the McDonald Omega (Ω) has not been calculated. Furthermore, it is also necessary to estimate the convergent validity using the Extracted Mean Variance (AVE).

It is good practice to perform a confirmatory factor analysis.  Is need to assess the validity of the constructs of the scales: HADS and SOC-3 (Confirmatory Factor Analysis: Relative Chi-Square, P; IFI; GFI; AGFI; CFI; RMSEA…)

The authors should explain more clearly all questionnaires.

Results:

The results should be presented more clearly.

The ANOVA analysis requires key assumptions: independent variables, univariate normality (Test Kolmogorov-Smirnov) and homoscedasticity, the assumption of equality of variances (Test Levene).

A linear regression analysis (LRA) was applied. LRA requires key assumptions: linear relationship, multivariate normality, homoscedasticity and finally, the linear regression assumes that there is little or no multicollinearity in the data. The explained variance (R2) in the mediation models is low (0.207 and 0.243)

Limitations:

Finally, the authors should consider another limitations.

The participants were not randomly assigned.

The explained variance (R2) in the mediation models is low (0.207 and 0.243)

Thank you.

Author Response

Reviewer 2

We would like to thank the reviewer for the time, effort and dedication to review our manuscript and suggest useful comments. We address each issue point-by-point below.

Thank you for the opportunity to review the manuscript entitled, "The comparison of students‘ anxiety, depression, suicidal risk and self-reported health before and during the COVID-19 pandemic”.

I believe this study investigated a topic relevant to the readers of “IJERPH”. The analysis of mental health among students of higher education who may be at risk of emotional or behavioral disorders has become a major focal point for researchers on a global scale, and it is now a priority for public health policies… The comparison of students‘ anxiety, depression, suicidal risk and self-reported health before and during the COVID-19 pandemic would significantly aid the proposals of improvement related to psychological orientation and counselling in this context, and it would also allow the definition of proceedings aimed to manage stress and anxiety in universities, especially in situations of crisis.

My opinion is that in the current form the manuscript does not have scientific quality to be published in IJERPH.

Structure of manuscript:

The authors should consider in your work the sections “Participants”, “Instruments”, “Procedure” and “Design”.

We agree. The Methods section was subdivided into “Participants”, “Design and procedure” and “Instruments”. Due to general reductions of data analyzed (we eliminated data from 2019 as suggested by Reviewer 4) our study became a simple cross-sectional survey. Thus, we believe that all the methodological data related to study design is now covered in the “Design and procedure” section.

Introduction:

The introduction is very simple, does not analyze in detail the variables under study and the relationship between variables. Neither is the variable “Sense of Coherence” analyzed. The introduction lacks clear objectives. The discussion seems to contain information (studies) that would have been more appropriate in the Introduction. The introduction must be rewritten.

We agree and have rewritten the introduction. Suicidal risk as well as the continuum of the symptomatology of anxiety and depression are discussed in the Introduction, while detail presentations of the study variables have been left in the Methods section (we believe that to be a more coherent approach). Now the introduction contains a paragraph specifically intended for a sense of coherence and its significance during the current pandemic, both in the general public as well as in a student population:

Various studies investigated factors (both modifiable and not) that may affect the impact that the pandemic has on students‘ mental well-being – most frequent variables assessed are sex, living conditions, family income stability, level of social support, having relatives infected with COVID-19, history of mental disorder, etc. [28–30]. However, only a few studies focused on a person’s inner resources to cope with new problems related to the COVID-19 crisis. For example, it was suggested that high sense of coherence, i.e. the ability to perceive stressful situations as understandable, manageable and meaningful, may be beneficial during the COVID-19 pandemic [31]. Various studies supported such assumptions as higher sense of coherence may serve as a protective factor for mental well-being and has been associated with higher life satisfaction during the health crisis [31–33]. Similarly, higher sense of coherence in university students was associated with better mental health and a health promoting lifestyle, yet these studies were done before the pandemic [34,35]. However, studies that assess students‘ sense of coherence and its impact on their mental well-being are lacking: to our knowledge, only one study investigated students‘ sense of coherence and it‘s mediating effect on mental health during the pandemic [36]

We have clarified the objectives of our study:

The aims of our study were to (1) evaluate students‘ levels of depression, anxiety and suicidal ideation and behavior and to (2) investigate variables associated with students‘ suicidality during the COVID-19 pandemic.

Participants:

The participants were not randomly assigned. The number of participants from various higher education institutions in Lithuania is not representative and directly affects external validity. The higher education institutions in Lithuania was chosen at random? The authors should explain more clearly how the participants have been selected for the study.

We agree that this is one of the biases of our study. The process of participant inclusion is now better defined under “2.2. Design and procedure”:

We used a non-probability sampling technique by distributing an online questionnaire from 31 January to 7 February 2021 during the second local wave of COVID-19 and a national lockdown in Lithuania. The online survey was distributed via social media groups, such as: “Medical students from X university/college”, “All students from X university/college“, “Students of Economics 2019”, “All students from X city” and similar. We searched for existing groups based on a list of institutions of higher education in Lithuania [38]. The number of members in these groups ranged from a dozen to more than a thousand. Students were kindly asked to complete the online anonymous survey and to share the questionnaire with their acquaintances through various social media channels. The latter technique was used to increase the reach and sample size of our survey. The survey took approximately 10 minutes to complete. We did not use any inclusion or exclusion criteria for our respondents, thus, every student who saw the invitation to complete the survey could have participated. An exception was exchange and foreign students who were unable to complete the questionnaire in Lithuanian. Because of the snowball sampling technique used to collect information we could not state the response rate of the survey.

We also added a sentence in Limitations:

Non-respondent bias may have also favored the inclusion of students who already suffer from mental health issues or are especially introspective towards their mental well-being and exclusion of healthy participants. This fact as well as a non-probability sampling technique decreased the external validity of our study

The general characteristics of the sample need be addressed in the material and methods section (participants).

We agree. The characteristics are now presented in Table 1 (under “2.1. Participants”).

Instruments:

In first place, the instruments must always display two important qualities: reliability and validity. Is need the Cronbach's alpha (α) and the McDonald Omega (Ω) has not been calculated. Furthermore, it is also necessary to estimate the convergent validity using the Extracted Mean Variance (AVE).

We agree. All three measures are now presented for the HADS scale. However, only measures of reliability were applicable for the SOC-3 as it is composed of only three questions and one factor.

HADS: In the current study, the reliability of the scale was acceptable (HADS-A: Cronbach’s α=0.84, McDonald’s ω=0.85, HADS-D: Cronbach’s α=0.74, McDonald’s ω=0.74, HADS (total): Cronbach’s α=0.86, McDonald’s ω=0.86). A confirmatory factor analysis indicated adequate construct validity (Table 2), while the estimate of convergent validity by the average variance extracted (AVE) was <0.50 (AVE (HADS-A)=0.454, AVE (HADS-D) =0.317) and composite reliability >0.70 (CR (HADS-A)=0.849 CR(HADS-D)=0.761)

SOC-3: In our study, the scale had optimal reliability (Cronbach’s α=0.84, McDonald’s ω=0.85).

It is good practice to perform a confirmatory factor analysis.  Is need to assess the validity of the constructs of the scales: HADS and SOC-3 (Confirmatory Factor Analysis: Relative Chi-Square, P; IFI; GFI; AGFI; CFI; RMSEA…)

We agree. Results of CFA for HADS are now presented in Table 2 (CFA for SOC-3 was not feasible as it is composed of a single factor). Further, an exploratory factor analysis for COVID-19 related items is presented in Table 3.

Table 2. Results of a confirmatory factor analysis for the HADS.

Characteristic

Value

χ2

466.57

df

76

p

<0.001

χ2/df

6.139

CFI

0.911

RFI

0.876

IFI

0.912

TLI

0.894

RMSEA

0.072

CFI – The Comparative Fit Index, df – degrees of freedom, IFI – Incremental Fit Index, RFI – the Relative Fit Index, RMSEA – The Root Mean Square Error of Approximation

Table 3. Results of an exploratory factor analysis (principal component analysis) after an oblique rotation (factor loadings >0.40 are highlighted) when questions related to the consequences of the COVID-19 pandemic were considered.

Factor

1 (“mental health”)

2 (“studies”)

3 (“socioeconomic”)

Variance explained

39.4%

12.7%

11.3%

COVID-19-related variable

More sadness

0.844

0.079

-0.003

More anxiety

0.843

0.034

-0.017

Worse physical health

0.694

0.062

-0.034

Worse study results

0.035

0.833

-0.079

Difficulties to concentrate

0.198

0.741

0.002

No comfortable place to study

-0.068

0.677

0.211

Lower income

-0.078

0.038

0.837

Worse career prospects

0.042

0.149

0.702

Worse interpersonal relationships

0.468

-0.160

0.473

The authors should explain more clearly all questionnaires.

We agree. The Methods section has been expanded to better present the measures used (please refer to subsections 2.3.1-2.3.5 for each instrument). Please note that the questionnaire is also provided as a supplement to the manuscript.

Results:

The results should be presented more clearly.

We agree, the results section has been rewritten (results of the sample before the pandemic have been removed as suggested by Reviewer 4). This section firstly presents correlational analysis, then proceeds with between-group comparisons and ends with results of binary regression models.

The ANOVA analysis requires key assumptions: independent variables, univariate normality (Test Kolmogorov-Smirnov) and homoscedasticity, the assumption of equality of variances (Test Levene).

We agree. After removal of results of the pre-pandemic survey (as suggested by Reviewer 4), we now do not report ANOVA results (however, the use of ANOVA is mentioned in the Methods section, but it yielded no statistically significant results to report):

The normality of variable distribution was assessed by the Kolmogorov-Smirnov test. Mann-Whitney U, Kruskal-Wallis (H) tests and Spearman’s correlation were used for be-tween-group (e.g., male vs. female, students from different fields of study) comparison and correlation analysis (between COVID-19-related items, HADS, SOC-3 and SRHS) for data of non-normal distribution as well as single-item Likert scale ordinal data. The Student’s t test or the one-way ANOVA were employed for normally-distributed data with equality of variances.

A linear regression analysis (LRA) was applied. LRA requires key assumptions: linear relationship, multivariate normality, homoscedasticity and finally, the linear regression assumes that there is little or no multicollinearity in the data. The explained variance (R2) in the mediation models is low (0.207 and 0.243)

We agree. However, we note that the linear regression model has been removed as suggested by Reviewer 4.

Limitations:

Finally, the authors should consider another limitations.

We agree. The section of study limitations has been expanded:

Further, all data was self-reported rather than collected by specialists in mental health through interviews or other means of objective evaluation. The intensity or risk factors of suicidal ideation and behavior were not assessed as well, thus, the interpretation of suicidal risk in our study should be cautious.

This fact as well as a non-probability sampling technique decreased the external validity of our study.

The participants were not randomly assigned.

We agree that this was an issue with the previously presented design. Now only cross-sectional results during the COVID-19 pandemic are presented (thus, randomization was not feasible because of the design of our study). A sentence has been added:

Non-respondent bias may have also favored the inclusion of students who already suffer from mental health issues or are especially introspective towards their mental well-being and exclusion of healthy participants. This fact as well as a non-probability sampling technique decreased the external validity of our study.

The explained variance (R2) in the mediation models is low (0.207 and 0.243)

We agree the linear regression model has been removed as suggested by Reviewer 4 as well as because of low R2.

Thank you.

Thank you again for reviewing our manuscript.

Reviewer 3 Report

The aim of this study is to estimate the rates of depression, anxiety, suicidal ideations among Lithuanian students and compare these rates before and during COVID pandemic. This study reflects changes in the COVID pandemic period and the relevance of mental health symptoms problem among Lithuanian students population.

Below are my comments on this manuscript.

Authors compared two samples of students -658 responders ( in 2019) and 1001 (in 2021). Although the samples are not very small, it is not clear how they were calculated and whether these samples reflect student population.

It is not clear on what basis the questions (COVID items) were chosen and what the purpose of them.  What was the purpose using SOC-3 in this study ? 

Evaluation of suicidal risk was assessed by asking questions about suicidal ideations, plans and previous attempts over a 12-month period. Neither intensity nor other factors are assessed - therefore it is not enough to draw conclusions about the risk of suicidal behavior.

Lots of repetitions in the results section; statistically insignificant results are presented. Figure 1 does not provide valuable information but only takes up space in the manuscript.

Author Response

Reviewer 3

We would like to thank the reviewer for the effort to review our manuscript and to suggest thoughtful comments. We address each issue point-by-point below.

The aim of this study is to estimate the rates of depression, anxiety, suicidal ideations among Lithuanian students and compare these rates before and during COVID pandemic. This study reflects changes in the COVID pandemic period and the relevance of mental health symptoms problem among Lithuanian students population.

Below are my comments on this manuscript.

Authors compared two samples of students -658 responders ( in 2019) and 1001 (in 2021). Although the samples are not very small, it is not clear how they were calculated and whether these samples reflect student population.

We agree that this should be explained better.

A minimum sample size of 300-350 individuals was calculated and is now mentioned in the Methods section (it was exceeded during the study):

The minimum required sample size for our study was calculated in G*Power for a one-way ANOVA of six groups (based on field of study) with α=0.05, 1-β=0.95 and f=0.25 and was n=324

The process of participant inclusion and how they reflect the population is reflected under “2.2. Procedure”:

We used a non-probability sampling technique by distributing an online questionnaire from 31 January to 7 February 2021 during the second local wave of COVID-19 and a national lockdown in Lithuania. The online survey was distributed via social media groups, such as: “Medical students from X university/college”, “All students from X university/college“, “Students of Economics 2019”, “All students from X city” and similar. We searched for existing groups based on a list of institutions of higher education in Lithuania [38]. The number of members in these groups ranged from a dozen to more than a thousand. Students were kindly asked to complete the online anonymous survey and to share the questionnaire with their acquaintances through various social media channels. The latter technique was used to increase the reach and sample size of our survey. The survey took approximately 10 minutes to complete. We did not use any inclusion or exclusion criteria for our respondents, thus, every student who saw the invitation to complete the survey could have participated. An exception was exchange and foreign students who were unable to complete the questionnaire in Lithuanian. Because of the snowball sampling technique used to collect information we could not state the response rate of the survey.

It is not clear on what basis the questions (COVID items) were chosen and what the purpose of them.  What was the purpose using SOC-3 in this study ?

We now better explain the inclusion of COVID-19 items, which represent three domains of interest (an exploratory factor analysis was performed to better show the latent structure of these questions in our sample):

Finally, students had to evaluate their experiences during the COVID-19 pandemic. Nine statements were proposed by the authors: 1) “My physical health got worse”, 2) “I felt more anxiety than usually”, 3) “I felt more sadness than usually”, 4) “My academic performance got worse”, 5) “I had no comfortable place to study”, 6) “I had difficulties focusing on my studies”, 7) “My career prospects got worse”, 8) “My personal relationships got worse”, 9) “My income became lower”. Each statement was rated on a 5-item Likert scale, where 1 point means total disagreement and 5 points means total agreement. We did not utilize any validated assessment tools to evaluate pandemic-associated changes in students’ life because of a lack of such instruments in the scientific community at that time. Therefore, statements were considered ad hoc to investigate three main domains of interest: 1) changes in mental health, 2) changes in professional (student) activities and 3) socioeconomic consequences of the pandemic. If combined, the reliability of these questions was adequate (Cronbach’s α=0.80, McDonald’s ω=0.80) and an exploratory factor analysis suggested that the three domains explained 63.4% of the variance (Kaiser-Meyer-Olkin measure=0.827), Table 3.

The inclusion of the SOC-3 scale was to include an intrinsic feature of coping with stressful issues (please refer to the second last paragraph of the introduction). This has been better defined under “2.3.2. The SOC-3” and the following sentence:

This instrument was included to investigate whether an intrinsic mechanism of resilience may be related to mental health and self-reported issues during the COVID-19 pandemic.

A more extensive explanation of our choice for this variable (SOC-3) is now given in the Introduction as well:

Various studies investigated factors (both modifiable and not) that may affect the impact that the pandemic has on students‘ mental well-being – most frequent variables assessed are sex, living conditions, family income stability, level of social support, having relatives infected with COVID-19, history of mental disorder, etc. [28–30]. However, only a few studies focused on a person’s inner resources to cope with new problems related to the COVID-19 crisis. For example, it was suggested that high sense of coherence, i.e. the ability to perceive stressful situations as understandable, manageable and meaningful, may be beneficial during the COVID-19 pandemic [31]. Various studies supported such assumptions as higher sense of coherence may serve as a protective factor for mental well-being and has been associated with higher life satisfaction during the health crisis [31–33]. Similarly, higher sense of coherence in university students was associated with better mental health and a health promoting lifestyle, yet these studies were done before the pandemic [34,35]. However, studies that assess students‘ sense of coherence and its impact on their mental well-being are lacking: to our knowledge, only one study investigated students‘ sense of coherence and it‘s mediating effect on mental health during the pandemic [36].

Evaluation of suicidal risk was assessed by asking questions about suicidal ideations, plans and previous attempts over a 12-month period. Neither intensity nor other factors are assessed - therefore it is not enough to draw conclusions about the risk of suicidal behavior.

We agree that we cannot draw firm conclusions on various features of suicidality from this survey. A sentence has been added to Limitations:

The intensity or risk factors of suicidal ideation and behavior were not assessed as well, thus, the interpretation of suicidal risk in our study should be cautious.

Lots of repetitions in the results section; statistically insignificant results are presented. Figure 1 does not provide valuable information but only takes up space in the manuscript.

We agree. The results section was simplified and rewritten, Figure 1 was deleted. Only significant results are now presented.

Reviewer 4 Report

The topic of research is of great interest. The suicide rate in young people has become a social alarm and in this era in which we live with the uncertainty of a pandemic, analyzing the factors involved is of great relevance. However, the research questions are not sufficiently clearly stated.

- Title: The title is confusing since it implies that the evolution of suicide rates and other variables before and after COVID will be analyzed, which requires a longitudinal study. However, the authors have analyzed two different samples evaluated at two different points in time.

- Abstract: It is also confusing for the same reason. What do the authors mean by normal circumstances?

-The introduction could be restructured according to different aspects:

  1. a) Distinguish between mental disorders and depressive or anxious symptomatology.
  2. b) Why are mental disorders common in young people?
  3. c) Is it the mental disorder that leads to emotional or social problems or is it these problems that cause the disorder? Please, to elaborate this section taking into account the possible bidirectionality (top-down vs bottom-up models)
  4. d) Why are students considered a vulnerable group to experience mental health problems?
  5. e) It is suggested that the introduction section be reordered. The authors begin by discussing mental health problems but it is not until the penultimate paragraph that they report on the incidence and prevalence of mental disorders.
  6. f) The authors point out that studies comparing the mental health of young people before and after pandemic are scarce and in any case are cross-sectional, but they also propose a similar design. What does their study contribute?
  7. g) In my view, previous studies relating anxiety, depression, mental health, sense of coherence and suicide to COVID needs to be addressed in more detail. There is a lot of information on this subject:

Stroud, I., & Gutman, L. M. (2021). Longitudinal changes in the mental health of UK young male and female adults during the COVID-19 pandemic. Psychiatry Research, 303, 114074.

Kwong, A. S., Pearson, R. M., Smith, D., Northstone, K., Lawlor, D. A., & Timpson, N. J. (2020). Longitudinal evidence for persistent anxiety in young adults through COVID-19 restrictions. Wellcome Open Research, 5(195), 195.

Kwong, A. S., Pearson, R. M., Adams, M. J., Northstone, K., Tilling, K., Smith, D., ... & Timpson, N. J. (2021). Mental health before and during the COVID-19 pandemic in two longitudinal UK population cohorts. The British Journal of Psychiatry, 218(6), 334-343.

Czenczek-Lewandowska, E., Wyszyńska, J., Leszczak, J., Baran, J., Weres, A., Mazur, A., & Lewandowski, B. (2021). Health behaviours of young adults during the outbreak of the Covid-19 pandemic–a longitudinal study. BMC public health21(1), 1-10.

Bendau, A., Kunas, S. L., Wyka, S., Petzold, M. B., Plag, J., Asselmann, E., & Ströhle, A. (2021). Longitudinal changes of anxiety and depressive symptoms during the COVID-19 pandemic in Germany: The role of pre-existing anxiety, depressive, and other mental disorders. Journal of anxiety disorders79, 102377.

Hawes, M. T., Szenczy, A. K., Klein, D. N., Hajcak, G., & Nelson, B. D. (2021). Increases in depression and anxiety symptoms in adolescents and young adults during the COVID-19 pandemic. Psychological Medicine, 1-9.

Varga, T. V., Bu, F., Dissing, A. S., Elsenburg, L. K., Bustamante, J. J. H., Matta, J., ... & Rod, N. H. (2021). Loneliness, worries, anxiety, and precautionary behaviours in response to the COVID-19 pandemic: a longitudinal analysis of 200,000 Western and Northern Europeans. The Lancet Regional Health-Europe2, 100020.

Amendola, S., von Wyl, A., Volken, T., Zysset, A., Huber, M., & Dratva, J. (2021). A longitudinal study on generalized anxiety among university students during the first wave of the COVID-19 pandemic in Switzerland. Frontiers in Psychology12.

ZHU, S., Zhuang, Y., Lee, P., & Ching, W. W. (2021). The changes of suicidal ideation status among young people in Hong Kong during COVID-19: A longitudinal survey. PsyArXiv. June1.

Wang, D., Ross, B., Zhou, X., Meng, D., Zhu, Z., Zhao, J., ... & Liu, X. (2021). Sleep disturbance predicts suicidal ideation during COVID-19 pandemic: A two-wave longitudinal survey. Journal of Psychiatric Research143, 350-356.

Hyland, P., Rochford, S., Munnelly, A., Dodd, P., Fox, R., Vallières, F., ... & Murphy, J. (2021). Predicting risk along the suicidality continuum: A longitudinal, nationally representative study of the Irish population during the COVID‐19 pandemic. Suicide and Life‐Threatening Behavior.

Czeisler, M. É., Wiley, J. F., Facer-Childs, E. R., Robbins, R., Weaver, M. D., Barger, L. K., ... & Rajaratnam, S. M. (2021). Mental health, substance use, and suicidal ideation during a prolonged COVID-19–related lockdown in a region with low SARS-CoV-2 prevalence. Journal of Psychiatric Research.

  1. h) The introduction is generally poor, it is necessary to address in detail how the pandemic impacts the different variables under study. The research gaps were not well addressed.

- The purpose of the research is not clear.

- In the Method section, it remains to include the participant sections and procedure. In the procedure section, it is necessary to explain how the sample was collected, if there were subjects who responded in both instrument passes, or how the quality of the collected data was controlled.

- It is suggested to rewrite the “Study design” section to distinguish between the procedure and create an instrument section. In addition, in this section it should be clarified that the questionnaires were applied before and after the COVID-19 to two different samples using a cross-sectional design. However, from my point of view, the main deficiency of the study is precisely the comparison of two different samples to explain how the assessed symptoms change before and during the COVID-19 pandemic. It would be advisable for the authors to focus exclusively on the results they obtained during COVID-19 and eliminate everything related to the first sample. Both samples are not comparable since they differ in gender, type of studies and people with whom they live, so the differences found in the variables of interest before and after the COVID-19 may be due to the characteristics of the sample and not to the COVID-19.

- The authors need to improve the description of the instruments: the response scale, the internal consistency of each original instrument and that obtained in this study are not reported... The authors also do not indicate how many questions the self-perceived health status or the suicidal thinking instrument had or how the response scale was. They also do not report how many questions were asked in relation to the COVID-19 or the response scale, nor how the instrument is used, nor its internal consistency. Please, add items example if possible. As the study is described, it can hardly be replicated.

- In the data analysis section, the analyzes performed are cited but it is not specified what each of them were used for.

- The results are presented in a confusing way and it is difficult to follow what is intended to be analyzed. It is suggested to order the results starting with the descriptive analyzes, continue with the correlational analyzes, the differences of means and, finally, the logistic regression. However, it will have to be reviewed based on the previous suggestions, focus only on the second sample and clarify the objective of the investigation, which are the predictor variables and which are the criterion variable?

- In the same line, it is recommended to review the presentation of the results in the tables. Table 1 should contain only descriptive statistics, clarifying if it refers to the number of subjects or percentages. It is also necessary to clarify what HADS-D >10 and HADS-A >10 refer to. The mean differences could appear in a separate table, including mean and standard deviation values. In addition, it is suggested to clarify when and why t, U or Anova is used.

- In the table 2, I seem to have detected some misprints when the authors include a value in the constant and it appears as significant. Additionally, the Wald statistic is missing and perhaps the significance could be represented with asterisks instead of dedicating a column to these data.

- It is recommended to include in the logistic regression the variables that were measured in relation to COVID-19 along with the remaining variables to predict suicide risk. This analysis makes more sense than linear regression models for predicting anxiety and depression. The authors have to clarify which are the predictor variables and which are the outcome variable.

- The description of the results is difficult to understand, for example because the authors use U to analyze gender differences in depression and use t for anxiety?

- Figure 1 is of low quality and the description should appear at the beginning of the results. In addition, when these variables are used, they must be described in the same way, the denomination does not coincide with the one presented in table 3.

- It would be appreciated if the authors accept my suggestions and the discussion section can also be modified accordingly.

- In the discussion section, to explain what you mean by which medical studies are more stressful. To explain the results in relation to the studies of humanities and social sciences.

- Reorder the discussion to explain the results together without distinguishing the different variables by subheadings.

In sum, if the authors are able to attend to the different suggestions, the manuscript could be reconsidered.

Author Response

Reviewer 4

We thank the reviewer for dedicating time and giving thoughtful arguments to improve our manuscript. We address each point below. Please note that all data relating to the pre-COVID-19 sample has been removed as you suggested.

The topic of research is of great interest. The suicide rate in young people has become a social alarm and in this era in which we live with the uncertainty of a pandemic, analyzing the factors involved is of great relevance. However, the research questions are not sufficiently clearly stated.

- Title: The title is confusing since it implies that the evolution of suicide rates and other variables before and after COVID will be analyzed, which requires a longitudinal study. However, the authors have analyzed two different samples evaluated at two different points in time.

We agree. The title has been changed to reflect the new structure of the manuscript (analysis of only data from 2021):

Mental health among higher education students during the COVID-19 pandemic: a cross-sectional survey from Lithuania

- Abstract: It is also confusing for the same reason. What do the authors mean by normal circumstances?

We agree and have rewritten the abstract to represent the cross-sectional design. The term “normal circumstances” was too abstract and has been removed.

-The introduction could be restructured according to different aspects:

Please refer to individual changes in text below each point and a general answer below numbered points.

  1. a) Distinguish between mental disorders and depressive or anxious symptomatology.

For example, a report by the WHO World Mental Health Surveys International College Student initiative estimated that the prevalence of mental disorders in a sample of nearly 14000 students was around 30% [3]. However, students’ mental distress may be even higher: as depressive and anxious symptoms form a continuum and range from none to severe, students may express such levels of anxiety and depression that may not be considered to be clinically significant [4–6].

  1. b) Why are mental disorders common in young people?

High prevalence of mental health disturbances as well as suicidal ideation and behavior among studying young adults is mainly explained by difficulties related to transition into adulthood [12,13]. In addition to stress posed by academic pressure, students experience future-projected uncertainties related to later marriage and childbirth, a later start of their careers [14]. Finally, many students who express high levels of mental distress do not reach for mental health counseling services due to fear of discrimination, concerns about the cost of such services or even not knowing that such services exist [15–17].

  1. c) Is it the mental disorder that leads to emotional or social problems or is it these problems that cause the disorder? Please, to elaborate this section taking into account the possible bidirectionality (top-down vs bottom-up models)

Mental distress (both clinically diagnosed and sub-syndromic factors) is considered to negatively influence academic performance and social well-being of the youth (however, due to bidirectional relationships an opposite pathway is also possible) [7].

  1. d) Why are students considered a vulnerable group to experience mental health problems?

High prevalence of mental health disturbances as well as suicidal ideation and behavior among studying young adults is mainly explained by difficulties related to transition into adulthood [12,13]. In addition to stress posed by academic pressure, students experience future-projected uncertainties related to later marriage and childbirth, a later start of their careers [14]. Finally, many students who express high levels of mental distress do not reach for mental health counseling services due to fear of discrimination, concerns about the cost of such services or even not knowing that such services exist [15–17].

Alongside struggles in maintaining personal relationships due to general social distancing measures and restrictions, students faced difficulties in maintaining their attention during the learning process [18]. Moreover, students experienced financial instability due to lost on-campus jobs [19]. Disrupted research projects and internships impaired students‘ competitiveness on the future job market [19]. Thus, young adults are considered to be one of the most vulnerable groups in terms of mental health issues during the COVID-19 crisis [20–22].

  1. e) It is suggested that the introduction section be reordered. The authors begin by discussing mental health problems but it is not until the penultimate paragraph that they report on the incidence and prevalence of mental disorders.

We agree. The section is now reordered accordingly.

  1. f) The authors point out that studies comparing the mental health of young people before and after pandemic are scarce and in any case are cross-sectional, but they also propose a similar design. What does their study contribute?

We provide arguments that our study provides new data that has not been covered by other studies:

However, the relationship between the current pandemic and suicidal ideation and behavior among young adults remains unclear: prospective studies from Hong Kong and Lithuania similarly highlighted that even though some individuals became suicidal during the pandemic, the majority remained non-suicidal or even recovered from suicidality [26,27].

However, studies that assess students‘ sense of coherence and its impact on their mental well-being are lacking: to our knowledge, only one study investigated students‘ sense of coherence and its mediating effect on mental health during the pandemic [36].

  1. g) In my view, previous studies relating anxiety, depression, mental health, sense of coherence and suicide to COVID needs to be addressed in more detail. There is a lot of information on this subject:

Stroud, I., & Gutman, L. M. (2021). Longitudinal changes in the mental health of UK young male and female adults during the COVID-19 pandemic. Psychiatry Research, 303, 114074.

Kwong, A. S., Pearson, R. M., Smith, D., Northstone, K., Lawlor, D. A., & Timpson, N. J. (2020). Longitudinal evidence for persistent anxiety in young adults through COVID-19 restrictions. Wellcome Open Research, 5(195), 195.

Kwong, A. S., Pearson, R. M., Adams, M. J., Northstone, K., Tilling, K., Smith, D., ... & Timpson, N. J. (2021). Mental health before and during the COVID-19 pandemic in two longitudinal UK population cohorts. The British Journal of Psychiatry, 218(6), 334-343.

Czenczek-Lewandowska, E., Wyszyńska, J., Leszczak, J., Baran, J., Weres, A., Mazur, A., & Lewandowski, B. (2021). Health behaviours of young adults during the outbreak of the Covid-19 pandemic–a longitudinal study. BMC public health, 21(1), 1-10.

Bendau, A., Kunas, S. L., Wyka, S., Petzold, M. B., Plag, J., Asselmann, E., & Ströhle, A. (2021). Longitudinal changes of anxiety and depressive symptoms during the COVID-19 pandemic in Germany: The role of pre-existing anxiety, depressive, and other mental disorders. Journal of anxiety disorders, 79, 102377.

Hawes, M. T., Szenczy, A. K., Klein, D. N., Hajcak, G., & Nelson, B. D. (2021). Increases in depression and anxiety symptoms in adolescents and young adults during the COVID-19 pandemic. Psychological Medicine, 1-9.

Varga, T. V., Bu, F., Dissing, A. S., Elsenburg, L. K., Bustamante, J. J. H., Matta, J., ... & Rod, N. H. (2021). Loneliness, worries, anxiety, and precautionary behaviours in response to the COVID-19 pandemic: a longitudinal analysis of 200,000 Western and Northern Europeans. The Lancet Regional Health-Europe, 2, 100020.

Amendola, S., von Wyl, A., Volken, T., Zysset, A., Huber, M., & Dratva, J. (2021). A longitudinal study on generalized anxiety among university students during the first wave of the COVID-19 pandemic in Switzerland. Frontiers in Psychology, 12.

ZHU, S., Zhuang, Y., Lee, P., & Ching, W. W. (2021). The changes of suicidal ideation status among young people in Hong Kong during COVID-19: A longitudinal survey. PsyArXiv. June, 1.

Wang, D., Ross, B., Zhou, X., Meng, D., Zhu, Z., Zhao, J., ... & Liu, X. (2021). Sleep disturbance predicts suicidal ideation during COVID-19 pandemic: A two-wave longitudinal survey. Journal of Psychiatric Research, 143, 350-356.

Hyland, P., Rochford, S., Munnelly, A., Dodd, P., Fox, R., Vallières, F., ... & Murphy, J. (2021). Predicting risk along the suicidality continuum: A longitudinal, nationally representative study of the Irish population during the COVID‐19 pandemic. Suicide and Life‐Threatening Behavior.

Czeisler, M. É., Wiley, J. F., Facer-Childs, E. R., Robbins, R., Weaver, M. D., Barger, L. K., ... & Rajaratnam, S. M. (2021). Mental health, substance use, and suicidal ideation during a prolonged COVID-19–related lockdown in a region with low SARS-CoV-2 prevalence. Journal of Psychiatric Research.

  1. h) The introduction is generally poor, it is necessary to address in detail how the pandemic impacts the different variables under study. The research gaps were not well addressed.

We have rewritten the introduction section. As suggested, we focused on the problem that young adults (a considerable part of them studying at higher education institutions. i.e., universities and colleges) are especially vulnerable to mental distress. We discussed the reasons for such mental distress among them (shift towards adulthood, new adult-like responsibilities (close relationships, financial independence etc.), academic pressure) and provided actual data that indicate high prevalence of mental health issues among students, i.e. depression, anxiety, suicidal ideation. Further, we have addressed how the current pandemic specifically affects students (difficulties of focusing on studies due to remote learning, lost on-campus jobs etc.).

We have analyzed all the articles provided and were grateful to get additional insights on the current problem of mental health among young adults during the pandemic. We have cited some of them in the introduction. Further, we found additional useful articles that were cited by those articles provided.

The main research gap we intended to highlight is that there is a lack of studies analyzing the sense of coherence among higher education students specifically during the pandemic. Thus, it is not clear how inner resilient resources of students may mediate the negative effects of pandemic-associated stressors on students’ mental health outcomes (depression, anxiety, suicidal ideation).

- The purpose of the research is not clear.

We have clarified the objectives of our study:

The aims of our study were to (1) evaluate students‘ levels of depression, anxiety and suicidal ideation and behavior and to (2) investigate variables associated with students‘ suicidality during the COVID-19 pandemic.

- In the Method section, it remains to include the participant sections and procedure. In the procedure section, it is necessary to explain how the sample was collected, if there were subjects who responded in both instrument passes, or how the quality of the collected data was controlled.

We agree. The process of participant inclusion and how they reflect the population is reflected under “2.2. Design and procedure”:

We used a non-probability sampling technique by distributing an online questionnaire from 31 January to 7 February 2021 during the second local wave of COVID-19 and a national lockdown in Lithuania. The online survey was distributed via social media groups, such as: “Medical students from X university/college”, “All students from X university/college“, “Students of Economics 2019”, “All students from X city” and similar. We searched for existing groups based on a list of institutions of higher education in Lithuania [38]. The number of members in these groups ranged from a dozen to more than a thousand. Students were kindly asked to complete the online anonymous survey and to share the questionnaire with their acquaintances through various social media channels. The latter technique was used to increase the reach and sample size of our survey. The survey took approximately 10 minutes to complete. We did not use any inclusion or exclusion criteria for our respondents, thus, every student who saw the invitation to complete the survey could have participated. An exception was exchange and foreign students who were unable to complete the questionnaire in Lithuanian. Because of the snowball sampling technique used to collect information we could not state the response rate of the survey.

- It is suggested to rewrite the “Study design” section to distinguish between the procedure and create an instrument section. In addition, in this section it should be clarified that the questionnaires were applied before and after the COVID-19 to two different samples using a cross-sectional design. However, from my point of view, the main deficiency of the study is precisely the comparison of two different samples to explain how the assessed symptoms change before and during the COVID-19 pandemic. It would be advisable for the authors to focus exclusively on the results they obtained during COVID-19 and eliminate everything related to the first sample. Both samples are not comparable since they differ in gender, type of studies and people with whom they live, so the differences found in the variables of interest before and after the COVID-19 may be due to the characteristics of the sample and not to the COVID-19.

We agree on the possible bias of comparing such different samples. Thus, we have eliminated the data of 2019 and, as suggested, focused on the students from the sample of 2021.

- The authors need to improve the description of the instruments: the response scale, the internal consistency of each original instrument and that obtained in this study are not reported... The authors also do not indicate how many questions the self-perceived health status or the suicidal thinking instrument had or how the response scale was. They also do not report how many questions were asked in relation to the COVID-19 or the response scale, nor how the instrument is used, nor its internal consistency. Please, add items example if possible. As the study is described, it can hardly be replicated.

We have improved the description of the instruments (every instrument is now discussed in a separate subsection in the methods section). We have provided additional data required for the psychometric description of the instruments. As suggested, we have added an English version of our questionnaire used in the survey - it is now provided as supplementary material.

- In the data analysis section, the analyzes performed are cited but it is not specified what each of them were used for.

We agree and now emphasize that the tests were used for between group analysis (and provide examples of the groups compared):

The normality of variable distribution was assessed by the Kolmogorov-Smirnov test. Mann-Whitney U, Kruskal-Wallis (H) tests and Spearman’s correlation were used for between-group (e.g., male vs. female, students from different fields of study) comparison and correlation analysis (between COVID-19-related items, HADS, SOC-3 and SRHS) for data of non-normal distribution as well as single-item Likert scale ordinal data. The Student’s t test or the one-way ANOVA were employed for normally-distributed data with equality of variances. Binary logistic regression modeling was used to estimate variables that are associated with suicidality.

- The results are presented in a confusing way and it is difficult to follow what is intended to be analyzed. It is suggested to order the results starting with the descriptive analyzes, continue with the correlational analyzes, the differences of means and, finally, the logistic regression. However, it will have to be reviewed based on the previous suggestions, focus only on the second sample and clarify the objective of the investigation, which are the predictor variables and which are the criterion variable?

We agree. Results are now restructured and start with correlational analyzes (descriptives are presented in the methods section under “Participants” as suggested by Reviewer 2), then proceed with between-group comparisons and end with regression. It is now emphasized in Table 5 which are the dependent variables and which are the independent ones.

- In the same line, it is recommended to review the presentation of the results in the tables. Table 1 should contain only descriptive statistics, clarifying if it refers to the number of subjects or percentages. It is also necessary to clarify what HADS-D >10 and HADS-A >10 refer to. The mean differences could appear in a separate table, including mean and standard deviation values. In addition, it is suggested to clarify when and why t, U or Anova is used.

We agree. As data regarding the pre-COVID-19 sample is now removed, only descriptive statistics of the newer sample are presented in the first table. HADS-A/D>10 refers to cut-off values used to determine clinically significant levels for symptoms of depression and anxiety (it is now clarified in “2.3.1. The HADS” section and presented in the results section, not Table 1). The use of statistical tests is better explained (their use was based on the normality of the variable):

The normality of variable distribution was assessed by the Kolmogorov-Smirnov test. Mann-Whitney U, Kruskal-Wallis (H) tests and Spearman’s correlation were used for between-group (e.g., male vs. female, students from different fields of study) comparison and correlation analysis (between COVID-19-related items, HADS, SOC-3 and SRHS) for data of non-normal distribution as well as single-item Likert scale ordinal data. The Student’s t test, the one-way ANOVA were employed for normally-distributed data with equality of variances

- In the table 2, I seem to have detected some misprints when the authors include a value in the constant and it appears as significant. Additionally, the Wald statistic is missing and perhaps the significance could be represented with asterisks instead of dedicating a column to these data.

We enter only independent variables from our study in the models and the value of the Intercept is determined by the statistical package used - it is statistically significant in only one of the models (Model 2). The Wald statistic is now provided in the table and p values are represented with asterisks, as suggested.

- It is recommended to include in the logistic regression the variables that were measured in relation to COVID-19 along with the remaining variables to predict suicide risk. This analysis makes more sense than linear regression models for predicting anxiety and depression. The authors have to clarify which are the predictor variables and which are the outcome variable.

We agree. COVID-19-related items are now included in the models. The linear regression model has been removed. It is now better emphasized in Table 5 which are the dependent variables and which are the independent ones

- The description of the results is difficult to understand, for example because the authors use U to analyze gender differences in depression and use t for anxiety?

We agree. The results section has been simplified. Please refer to the answer above for information regarding the use of statistical tests (their use was based on the normality of the variables).

- Figure 1 is of low quality and the description should appear at the beginning of the results. In addition, when these variables are used, they must be described in the same way, the denomination does not coincide with the one presented in table 3.

We agree. Figure 1 has been removed as suggested by Reviewer 3. The denomination of the variables is now the same throughout the manuscript.

- It would be appreciated if the authors accept my suggestions and the discussion section can also be modified accordingly.

We thank you again for the suggestions and have modified the discussion to suit them better.

- In the discussion section, to explain what you mean by which medical studies are more stressful. To explain the results in relation to the studies of humanities and social sciences.

We now elaborate on this topic. However, we would want to emphasize that students from all specialties should be treated equally (e.g., during public responses) as the HADS and suicidality-related questions yielded similar findings throughout the study-area subgroups:

During the COVID-19 pandemic, many studies investigated mental health of medical students as they typically show elevated levels of anxiety [83]. However, in contrast to the general view, we did not observe any significant differences in terms of depression, anxiety as well as suicidal ideation and behavior between medical students and students from other specialties. The Student Experience in the Research University (SERU) Consortium survey revealed similar results, as students of health sciences were not among those with highest prevalence of anxiety and depression [84]. The only difference observed in our study was the experience of worse career prospects due to the pandemic – it may be reasonable as medical students faced additional challenges related to remote studying and lack of ability to improve their clinical skills. However, our data highlights the importance to assess the mental health of all students of higher education equally rather than focus on one specialty

- Reorder the discussion to explain the results together without distinguishing the different variables by subheadings.

We agree. The discussion has been reorganized and the subheadings were removed.

In sum, if the authors are able to attend to the different suggestions, the manuscript could be reconsidered.

We thank you again for considering our article and providing insightful comments.

Round 2

Reviewer 1 Report

Thank you for addressing my comments.

Author Response

We thank the Reviewer for putting effort to review our manuscript for a second time.

Reviewer 2 Report

Thank you for the opportunity to review again the manuscript entitled, "The comparison of students‘ anxiety, depression, suicidal risk and self-reported health before and during the COVID-19 pandemic”; “Mental health among higher education students during the 2 COVID-19 pandemic: a cross-sectional survey from Lithuania”

The authors took into account my comments and they proceeded with the necessary revisions. I thank the authors their time and effort. My opinion is that in the current form the manuscript can be published

Thank you.

Author Response

We thank the Reviewer for a second round of revisions and are pleased to have addressed the previous comments.

Reviewer 3 Report

The authors  did substantial changes that significantly improved the manuscript.

Some my comments:

 Introduction sections is too long; 

Table 2: Confirmatory factor analysis of HADS is redundant information for this topic.

Author Response

The authors  did substantial changes that significantly improved the manuscript.

We thank the Reviewer for his time and effort to provide additional comments for a second round of revisions. Please find answers to each comment below.

Some my comments:

Introduction sections is too long;

We agree and now have removed the following paragraph:

For example, higher education students currently show increased levels of anxiety and depression [23–25]; data from Jordan indicated university students to be more anxious than healthcare professionals, even though the latter group seems to be most directly affected by the pandemic [23]. However, the relationship between the current pandemic and suicidal ideation and behavior among young adults remains unclear: prospective studies from Hong Kong and Lithuania similarly highlighted that even though some individuals showed increased risk of suicide during the pandemic, the majority remained non-suicidal or even recovered from suicidality [26,27].

Table 2: Confirmatory factor analysis of HADS is redundant information for this topic.

We agree and now have removed the CFA.

Reviewer 4 Report

The authors have made a considerable effort to respond to the various suggestions. The manuscript has improved remarkably, although there are still some questions to be resolved.

  • In the abstract, it is recommended to use the term mental health in the place of mental well-being since symptomatology has been evaluated, not well-being. Please correct throughout the manuscript.
  • In the abstract, clarify that High SRHS refers to a positive or negative assessment of your health status, depending on how it has been measured.
  • At the end of line 46 there is a missing “and” to connect the different reasons given for being more vulnerable to mental health problems.
  • On line 63, it is recommended to change the expression “some individuals became suicidal during the pandemic…” by “some individuals show increased risk of suicide during the pandemic…”
  • What do you mean by sex affects the impact of the pandemic?
  • On line 70, it would be necessary to include that “one of the person’s inner resources to cope with new problems” is the sense of coherence.
  • Now the aims of the study are clearer, but perhaps in the second objective we should include what type of variables it refers to that are associated with suicide risk.
  • In the description of participants, the authors comment that around 1% of the total population of university students participated in the study but they do not clarify how the sample was drawn: at random or convenience sample?; come from different educational centers or mostly from one area of the country?...
  • Similarly, it is not clear from the procedure how the data were obtained: how many institutions did the authors have access to? Was the link to the questionnaires only disseminated through social channels?
  • The variables that were included are now much clearer, mainly those related to suicide and health status (Table 1).
  • Specifically, what was the question that was asked to assess the health status?
  • On line 195, please, put table 3 in parentheses.
  • The authors report the minimum size of the sample to achieve an effect size, but although they report the followed procedure, it is not clear what variable they mean when they say " based on the question relating to the field of study”.
  • On line 216, separate the Spearman correlation analysis from the U and H analyzes that compare the groups based on a criterion variable with a dot.
  • According to APA standards, correlations can be reported to only two decimal places.
  • There is a typo in the note in Table 4 it should be HADS-D instead of HADS-B. The same thing happens in line 250 and in the note of TABLE 5. Furthermore, the negative sign that appears in front of the probability p should be removed.
  • In the comparison based on gender, the mean is reported for some variables but not for others. For example: physical health (Z=-2.01, p=0.045), increased anxiety (Z=-4.09, p<0.001) and depressiveness.
  • The authors comment that “Individuals whose family members were hospitalized be-240 cause of COVID-19 (n=39) more often indicated a worsening in study results (Z=-2.06, 241 p=0.039)”. Were they compared with individuals who did not have hospitalized relatives? In what variables were the differences found?
  • In the case of logistic regression “Only respondents who ever had suicidal thoughts (positive for general suicidal risk) were included” , what question or questions were selected to identify participants with suicidal risk? It is recommended in table 5, to put the N of each tested model.
  • Please include the full name of the GAD-7 questionnaire when it is cited for the first time (line 283).
  • In the discussion section, it is suggested to review the interpretation regarding high suicide rates. The authors argue that “We consider such measurements as a reflection of the poor general suicidal situation in Lithuania: according to the WHO, Lithuania ranks 7th among countries with the highest suicide rates in the world and first in Europe in 2019 [61]. Such assumption may be reasonable, as the onset age of suicidality is between 10 and 15 years [62].” but they do not explain why these rates may be due or what differentiates the country from others with lower rates.
  • Could the authors explain what the reported result on SHR might be due to? They argue that “during the pandemic, SRHS is more frequently treated as an endpoint than an independent variable [69–71]. Our study provides additional information that self-rated health status may be a significant variable associated with suicidal behavior among youth” but what could it be due to? That the state of health is associated with suicide attempts in young people who are usually in good health, needs a further explanation. For example: Could it be that young people think that their health status is diminished because they cannot perform physical activity?
  • The last four paragraphs of the discussion report the results regarding the correlations and differences of means between groups according to the sociodemographic variables. It is recommended to move these paragraphs to the beginning of the discussion so that they are in the same order in which the results were presented.
  • Excuse me, but I don't really understand the meaning of including the study carried out in China.

Author Response

The authors have made a considerable effort to respond to the various suggestions. The manuscript has improved remarkably, although there are still some questions to be resolved.

We thank the Reviewer for a continuous commitment to provide useful and timely comments to improve our manuscript. We address each point below.

  • In the abstract, it is recommended to use the term mental health in the place of mental well-being since symptomatology has been evaluated, not well-being. Please correct throughout the manuscript.

We agree and have corrected the term accordingly: We set out to investigate students’ mental health amidst..

  • In the abstract, clarify that High SRHS refers to a positive or negative assessment of your health status, depending on how it has been measured.

We have clarified the SRHS: High SRHS (higher score refers to more positive health status) was the only significant independent variable associated with less frequent suicidal attempts in the past year (p<0.01, OR=0.29 , 95% CI=0.12 to 0.66)

  • At the end of line 46 there is a missing “and” to connect the different reasons given for being more vulnerable to mental health problems.

The sentence has been adjusted to be more comprehensible:

Before: In addition to stress posed by academic pressure, students experience future-projected uncertainties related to later marriage and childbirth, a later start of their careers

After: In addition to stress posed by academic pressure, students experience future-projected uncertainties related to later marriage, childbirth and a later start of their careers

  • On line 63, it is recommended to change the expression “some individuals became suicidal during the pandemic…” by “some individuals show increased risk of suicide during the pandemic…”

We agree but note that the sentence has been removed to comply with the need to shorten the Introduction, as suggested by Reviewer 3.

  • What do you mean by sex affects the impact of the pandemic?

We had in mind that female students seem to be mentally negatively affected in a higher extent, compared to male counterparts, during the current pandemic. We have changed the term “sex“ into “female sex“ and added two articles that indicated female gender to be a risk factor. The directionality of each risk factor is now included:

Various studies investigated factors (both modifiable and not) that may affect the impact that the pandemic has on students‘ mental health – most frequent variables assessed are female sex, worse living conditions, family income instability, lower level of social support, having relatives infected with COVID-19, a history of mental disorder etc.

  • On line 70, it would be necessary to include that “one of the person’s inner resources to cope with new problems” is the sense of coherence.

We agree and have included an additional sentence: However, only a few studies focused on a person’s inner resources to cope with new problems related to the COVID-19 crisis. One of such resources is the sense of coherence.

  • Now the aims of the study are clearer, but perhaps in the second objective we should include what type of variables it refers to that are associated with suicide risk.

We have clarified the sentence by mentioning exact variables: depressiveness, anxiety, pandemic-associated experiences and sense of coherence.

  • In the description of participants, the authors comment that around 1% of the total population of university students participated in the study but they do not clarify how the sample was drawn: at random or convenience sample?; come from different educational centers or mostly from one area of the country?...

We agree that additional explanations are required.

We added information that the sample was acquired by means of a mixed methods of both convenience and snowball sampling: We used a non-probability (convenience and snowball) sampling technique…

We have added a sentence to clarify our expectations that the sample of students was heterogenous in terms of educational institutions they represent: We searched for existing groups based on a list of institutions of higher education in Lithuania; thus, students from different educational centers could have participated in the study [40]. 

We have added another sentence to clarify that our sample was drawn by trying to reach all social media groups that were found during that time: We reached-out to all social groups of students that were found in social media platforms“

  • Similarly, it is not clear from the procedure how the data were obtained: how many institutions did the authors have access to? Was the link to the questionnaires only disseminated through social channels?

We have added a sentence to clarify that only social media channels were used to reach students: “We did not use any other way to reach students“.

We believe, that the sentence added (“We reached-out to all social groups of students that were found in social media platforms “) now indicates more clearly that students from all educational centers registered in Lithuania (according to the Ministry of of Education, Science and Sport) could have participated. Regretfully, we still cannot specify the discrete number of institutions because of possible snowball technique-associated recruitments.

  • The variables that were included are now much clearer, mainly those related to suicide and health status (Table 1).

Thank you.

  • Specifically, what was the question that was asked to assess the health status?

Please find this and other questions provided in the supplementary data of our survey. The question that was asked: “How do you evaluate your health?” Answers: Very good/Good/Average/Poor/Very poor (rated 5, 4, 3, 2, 1, respectively)

  • On line 195, please, put table 3 in parentheses.

Thank  you, we have put table 3 in parentheses.

  • The authors report the minimum size of the sample to achieve an effect size, but although they report the followed procedure, it is not clear what variable they mean when they say " based on the question relating to the field of study”.

We agree that this may be clarified. The subgroups based on the field of study are now specified:

The minimum required sample size for our study was calculated in G*Power for a one-way ANOVA of six groups (based on the field of study, i.e. Biomedical, Physical, Social sciences, Humanities, Art, Technologies) with α=0.05, 1-β=0.95 and f=0.25 and was n=324.

  • On line 216, separate the Spearman correlation analysis from the U and H analyzes that compare the groups based on a criterion variable with a dot.

We agree and have edited those sentences: Mann-Whitney U, Kruskal-Wallis (H) tests were used for between-group (e.g., male vs. female, students from different fields of study) comparison. Spearman’s correlation was used for correlation analysis (between COVID-19-related items, HADS, SOC-3 and SRHS) for data of non-normal distribution as well as single-item Likert scale ordinal data.

  • According to APA standards, correlations can be reported to only two decimal places.

We agree and have edited the text and tables whenever correlations were presented.

  • There is a typo in the note in Table 4 it should be HADS-D instead of HADS-B. The same thing happens in line 250 and in the note of TABLE 5. Furthermore, the negative sign that appears in front of the probability p should be removed.

We agree and have corrected the typos.

  • In the comparison based on gender, the mean is reported for some variables but not for others. For example: physical health (Z=-2.01, p=0.045), increased anxiety (Z=-4.09, p<0.001) and depressiveness.

We agree. Means are now presented for all comparisons:

Female individuals had higher levels of anxiety (Mfemale=10.6 ±4.3 vs Mmale=8.7 ±4.3, Z=-5.72, p<0.001) and worse sense of coherence (Mfemale=2.7 ±1.3 vs Mmale=2.4 ±1.3, Z=-3.10, p=0.002). They more frequently reported worse physical health (Mfemale=3.5 ±1.2 vs Mmale=3.2 ±1.3, Z=-2.01, p=0.045), increased anxiety (Mfemale=3.7 ±1.6 vs Mmale=3.3 ±1.2, Z=-4.09, p<0.001) and depressiveness (Mfemale=3.7 ±1.2 vs Mmale=3.4 ±1.2, Z=-3.92, p<0.001) as consequences of the COVID-19 pandemic. Individuals whose family members were hospitalized because of COVID-19 (n=39) more often indicated a worsening in academic performance, compared to those who did not have hospitalized relatives (Mhospitalized=3.2 ±1.4 vs Mnot hospitalized=2.8 ±1.3, Z=-2.06, p=0.039).

  • The authors comment that “Individuals whose family members were hospitalized be-240 cause of COVID-19 (n=39) more often indicated a worsening in study results (Z=-2.06, 241 p=0.039)”. Were they compared with individuals who did not have hospitalized relatives? In what variables were the differences found?

Yes, they were compared with individuals who did not have hospitalized relatives. Difference was found in scoring of one of the statements regarding students‘ experiences during the pandemic (“My academic performance got worse”). We have clarified the sentence: “Individuals whose family members were hospitalized because of COVID-19 (n=39) more often indicated a worsening in academic performance, compared to those who did not have hospitalized relatives (Z=-2.06, p=0.039).“

  • In the case of logistic regression “Only respondents who ever had suicidal thoughts (positive for general suicidal risk) were included” , what question or questions were selected to identify participants with suicidal risk? It is recommended in table 5, to put the N of each tested model.

As mentioned in the Instruments subsection regarding evaluation of one‘s suicidal ideation and behavior (“Firstly, students reported whether they thought about committing suicide, planned or tried to commit suicide any time during their lifetime. If affirmative, they were considered to have risk for suicide and had to answer four other questions concerning symptoms of sadness and apathy as well as suicidal ideation and behavior during the past year.“), the first question was used to identify participants with general suicidal risk. We now clarify this in the caption of the table: Only respondents who ever had suicidal thoughts, plans or attempts (positive for general suicidal risk) were included in models 2 to 4.

Please find the number of individuals included in each model in respective headings of each model rather than the footnote of the table.

  • Please include the full name of the GAD-7 questionnaire when it is cited for the first time (line 283).

It is now included: This may explain the high rate of anxiety as many beforecited studies used the General-ized Anxiety Disorder scale-7 (GAD-7) questionnaire for detecting the disorder.

  • In the discussion section, it is suggested to review the interpretation regarding high suicide rates. The authors argue that “We consider such measurements as a reflection of the poor general suicidal situation in Lithuania: according to the WHO, Lithuania ranks 7th among countries with the highest suicide rates in the world and first in Europe in 2019 [61]. Such assumption may be reasonable, as the onset age of suicidality is between 10 and 15 years [62].” but they do not explain why these rates may be due or what differentiates the country from others with lower rates.

We agree that the topic is of great importance to our article. We now better explain the reasons for high suicide rates in Lithuania and then keep the idea of a possible relationship between the situation in Lithuania regarding suicide rates and high suicidality among youth:

This may be associated with significant socioeconomic shifts in the country during the last decades: the Lithuanian society emancipated from the Soviet regime and witnessed an economic boom until the global financial crisis in 2008 while having no effective national suicide prevention strategy. Despite recent improvements in mental health services, it may be reasonable to consider that students in Lithuania are highly exposed to suicidal behavior among peers, relatives as well as the general society, and the latter aspect seems to be predictive for increased suicidality.

  • Could the authors explain what the reported result on SHR might be due to? They argue that “during the pandemic, SRHS is more frequently treated as an endpoint than an independent variable [69–71]. Our study provides additional information that self-rated health status may be a significant variable associated with suicidal behavior among youth” but what could it be due to? That the state of health is associated with suicide attempts in young people who are usually in good health, needs a further explanation. For example: Could it be that young people think that their health status is diminished because they cannot perform physical activity?

We agree and have revised the paragraph related to SRHS and discussed the variety of factors that determine one‘s self-rated health. We also mentioned reduction of physical activity as a factor that may be significant in projecting suicidal-related outcomes in a students‘ population: Thus, our study provides additional information that self-rated health status may be a predictive factor for suicidal behavior among youth. However, the measurement of SRHS is determined by various physical, psychological as well as socio-demographic aspects. Therefore, it is difficult to clarify specific variables that may be the most relevant determinants of worse SRHS. We could only speculate that one of such factors may be a reduction of students’ physical activity due to the pandemic, as higher physical activity is related both to better self-rated health and lower suicidal ideation.

  • The last four paragraphs of the discussion report the results regarding the correlations and differences of means between groups according to the sociodemographic variables. It is recommended to move these paragraphs to the beginning of the discussion so that they are in the same order in which the results were presented.

We agree and have re-ordered these paragraphs.

  • Excuse me, but I don't really understand the meaning of including the study carried out in China.

We imagine the study in question to be Ren, Z et al., Frontiers in Psychology (2021), 12 (doi:10.3389/fpsyg.2021.641806). According to our previous revision, it was suggested that several additional factors (e.g., self- and class quarantine, routine mask wearing, temperature checking) may also contribute to mental health disturbances among students. Therefore, this study is to outline that such variables have also been shown to have some effect in the student population.